# Pig Coat Color Manipulation by *MC1R* Gene Editing

**DOI:** 10.3390/ijms231810356

**Published:** 2022-09-08

**Authors:** Haiwen Zhong, Jian Zhang, Cheng Tan, Junsong Shi, Jie Yang, Gengyuan Cai, Zhenfang Wu, Huaqiang Yang

**Affiliations:** 1National Engineering Research Center for Breeding Swine Industry, College of Animal Science, South China Agricultural University, Guangzhou 510642, China; 2Yunfu Branch Center of Guangdong Laboratory of Lingnan Modern Agricultural Science and Technology, Yunfu 527400, China

**Keywords:** coat color, CRISPR, gene editing, homologous recombination, *MC1R*, pig

## Abstract

Black coat color in pigs is determined by the dominant *E* allele at the *MC1R* locus. Through comparing *MC1R* gene sequences between recessive *e* and dominant *E^D1^* alleles, we identified four missense mutations that could affect MC1R protein function for eumelanin synthesis. With the aim of devising a genetic modification method for pig coat color manipulation, we mutated the *e* allele in the Duroc breed to the dominant *E^D1^* allele using CRISPR-mediated homologous recombination for the four mutation substitutions at the *MC1R* locus. The *MC1R*-modified Duroc pigs generated using the allele replacement strategy displayed uniform black coat color across the body. A genotyping assay showed that the *MC1R*-modified Duroc pigs had a heterozygous *E^D1^*/*e* allele at the *MC1R* locus; in addition, the pigs remained in the Duroc genetic background. Our work offers a gene editing method for pig coat color manipulation, which could value the culture of new pig varieties meeting the needs of diversified market.

## 1. Introduction

Domestic pigs have coat color diversity. The European domestic pig breeds usually have a white or red coat color, whereas most Chinese indigenous breeds have a black coat color. The differential coat color phenotypes are also a reflection of the different domestication origins between European and Asian pig breeds [1]. The domestication strategy of Chinese indigenous pig breeds confers a high fat accumulation in pigs, which improves the pork flavor [2,3,4]. Despite the smaller body size and weight as well as less meat production, the higher meat quality of Chinese indigenous breeds usually brings a better market value than the European lean breeds. Current Chinese customers favor consumption of meat of indigenous pigs that have a common representative trait of a black coat color.

Many genes have been identified that are involved in mammalian coat color development [5,6]. Among them, melanocortin receptor 1 (MC1R) encoded by *Extension* (*E*) coat color locus is a well-known regulator of melanogenesis, controlling the switch between two basic types of melanin: eumelanin (black/brown) and phaeomelanin (yellow/red) [7]. The active form of MC1R promotes eumelanin synthesis, producing a uniform black coat color, whereas inactive MC1R is associated with the eumelanin to phaeomelanin switch, resulting in red or yellow coat color [7,8]. In pigs, four *MC1R* variants corresponding to five distinct *E* alleles have been found to be associated with different coat color phenotypes ranging from recessive red to dominant black [9]. The *E* allele for black coat color is dominant over the *e* allele for red coat color [9]. The coat color genetics provides a molecular method to regulate coat color in mammals. One can readily alter coat colors of animals by using or disrupting the alleles controlling specific coat colors [10,11]. In this respect, black pigs can be generated by incorporating the *E* allele into the genome. We here present a gene editing strategy to replace the Duroc *e* allele with the *E^D1^* allele controlling the uniform black coat color, with the aim of cultivating a pig breed with high growth rate and meat production as well as a preferred coat color to meet the market demands.

## 2. Results and Discussion

The aim of the present study is to change the coat color of Duroc pigs from red to black. Through comparing the *MC1R* allele sequences among multiple pig breeds, several nucleotide sites with single nucleotide polymorphisms (SNPs) were found to cause missense mutations in *MC1R* gene. The four missense mutations between Duroc and indigenous black pigs are V95M, L102P, V164A, and T243A at amino acid positions of MC1R (Figure 1A). Of them, mutations at positions 95 and 102 are considered to be the specific common variations in Chinese indigenous black pigs [12]. V95M and L102P substitutions in the *E^D1^* allele may contribute to the active function of MC1R, thus accounting for the dominant black phenotype [13]. A164V and A243T are the specific missense mutations in the Duroc breed. A243T is believed to substantially disrupt MC1R function, resulting in a red coat color phenotype of Duroc pigs [9,13].

We set out to mutate all four missense SNPs in the *MC1R* coding region of Duroc pigs to the corresponding nucleotides in *MC1R* of Chinese indigenous black pigs (Figure 1A,B). We used a CRISPR-mediated homologous recombination (HR) approach to replace the Duroc *MC1R* allele with the donor plasmid harboring the four indicated SNPs (Figure 1B). As the black pig *MC1R E^D1^* allele is dominant over the Duroc recessive *e* allele, both heterozygous *E^D1^/e* and homozygous *E^D1^/E^D1^* genotypes can realize a black coat color phenotype. Through CRISPR and donor co-transfection into Duroc fetal fibroblasts, drug selection and enrichment, and cell colony genotyping, we obtained 40 cell colonies with predesigned substitutions in all four SNPs (with only one *e* allele being replaced with *E^D1^* allele of a homologous donor in most colonies), but no homozygous mutants at all four SNPs were screened out. The HR efficiency of precise replacement was 14.3% (40/280). This result indicates HR remains a low-efficiency event in primary fibroblast, even with the aid of CRISPR. Previous reports showed that CRISPR-mediated HR occurred in an efficiency range from 0 to 41.2% in pig fetal fibroblast with a very low rate in most cases, far less than the CRISPR-mediated knockout efficiency [14,15,16].

The positive fibroblast colonies were used as the nuclear donor for somatic cell nuclear transfer (SCNT). We totally constructed 1938 *MC1R*-modified embryos from 3 cell colonies and transferred them to 9 surrogates. Five surrogates were pregnant and maintained to term, and finally gave birth to 30 piglets on day 116 of gestation. Among them, 19 cloned piglets were born healthy and developed normally (Table 1). Analysis of the coat colors found that 17 were uniform solid dark black, 2 were light black, and 11 were red (Figure 2A,B). Recipients 19-937900, 20-694608, and 20-006896 received embryos derived from only one colony (#2-24) but gave birth to cloned pigs with different coat colors, implying that the colony contained a minor part of cells with unexpected genotypes. The *MC1R* genes of all cloned piglets were sequenced to investigate the relationship between *MC1R* genotypes and coat colors. The *E* alleles after HR were genotyped by combinations of three key SNPs (V95M, L102P, and T243A) affecting MC1R activity. The A164V mutation in the Duroc breed is observed in the functionally normal horse MC1R, thus has no effect on MC1R activity [9]. We found that all dark black pigs harbored at least one *E^D1^* allele with all four SNP substituted. Of the 17 dark black pigs, 8 had three SNPs substituted (V95M, L102P, and T243A) on one allele and one SNP substituted (V164A) on both alleles, and 9 had all four SNPs substituted on only one allele (Figure 2C and Table 2). Because AA164 is the closest SNP to the CRISPR-cleavage site (approximately 80 bp) among the four SNPs, AA164 has a much high HR rate than the other three SNPs. Both genotypes in dark black pigs after gene editing can be classified to *E^D1^/e*. The result clearly indicated that the *E^D1^* allele is dominant over the *e* allele, and presence of only one *E^D1^* allele in the Duroc genetic background can confer a uniform black coat color phenotype. In contrast, the cloned pigs with wild type red coat color all had none of substitutions in those SNP sites. Some red pigs harbored homozygous or heterozygous insertion or deletion (Indel) in the gRNA region that clearly had a knockout effect on the *MC1R* gene (Table 2). The same red coat color formed by either *MC1R e* allele expression or *MC1R* knockout suggests that the Duroc *e* allele is totally inactive in eumelanin synthesis.

It is worth noting that two cloned pigs (#8 and #27) displayed a light black coat color (Figure 2B and Figure 3A). *MC1R* genotyping showed that they both had at least one *e* allele replaced with an *E^D1^* allele with all four SNP substitutions. For the other *e* allele, #8 had two substituted SNPs (V164A and T243A, these mutations formed an *E^+^* allele) and #27 had no substitutions. They both had an intact *E^D1^* allele (Table 2 and Figure 3C). However, their black coat color seemed not completely dominant. We sequenced the whole *MC1R* coding and flanking non-coding regions of the two pigs and detected no extra mutations. We then isolated the mRNA from ear skin and measured the *MC1R* expression status. A qPCR analysis of *MC1R* mRNA level detected that #8 and #27 had a similar *MC1R* expression level with the wild-type red Durocs, whereas the dark black Durocs harboring the *E^D1^* allele had 3–5 times higher *MC1R* expression compared with the wild-type red Durocs (Figure 3B). These results imply that full dominance of *E^D1^* depends on its higher expression than that of *e*. An intact *E^D1^* but without sufficient expression level could not support a completely dominant black coat color. We further sequenced the *MC1R* cDNA of cloned pigs and found the light black pigs had an equal expression level for *E^D1^* and *e*, but the dark black pigs had a major expression for only *E^D1^*, although they are heterozygous for *E^D1^* and *e* in genotypes (Figure 3C). For the reason why the *E^D1^* are not expressed sufficiently in the two light black pigs, we suspected that the recombinant *E^D1^* allele has distal mutations or structural variations caused by the HR process, which impaired *E^D1^* expression. The detailed mechanism remains to be investigated further.

To investigate the genetic background of the clone piglets, we carried out an admixture analysis to estimate the proportion of ancestries using genome-wide SNP data of cloned pigs. The results showed that the cloned pigs with the dark black coat color had a nearly 100% Duroc genetic structure (Figure 4).

The solid black hair across the body of cloned pigs demonstrates that only one *E^D1^* allele introduction can fully confer the black coat color trait into the Duroc genetic background, further confirming that the red coat color in Duroc is a recessive trait compared to the black coat color. In this regard, transgenic expression of foreign *MC1R* with the *E^D1^* genotype can also achieve such coat color switching purposes. However, the transgenesis strategy could bring an inconsistent gene expression level because of the different transgene copies and integrated locations in the host genome, which may influence the coat color switching effect in transgenic pigs [17,18]. This point has been demonstrated in the light black pigs generated in our work. Although an intact *E^D1^* is incorporated into the genome in the light black pigs, a low expression level of *E^D1^* still cannot support a fully dominant black coat color. In addition, the transgenesis strategy is disadvantageous for subsequent pig breeding programs to culture the blackish lean pig breeds, as the transgene integration cannot be controlled to keep an identical genotype in different cells and pigs used for transgenesis [17,18]. Our HR-mediated point mutation strategy facilitates breeding of a new pig variety with black coat color and high meat yield.

Considering that multiple genetic loci can control coat color trait, the present *MC1R* manipulation strategy may not work well in some pig breeds with complex coat colors or patterns. For example, many European white pigs such as Large White and Landrace harbor the *dominant white* (*I*) locus, the molecular basis of which is duplication of the *KIT* gene [19]. *Dominant white* can mask other coat colors by allele dominance. The dominance relationship among these coat color gene loci can be reflected in pig crossbreeding practice. Black pig and Duroc crossbred usually has the uniform black coat color, implying a “black dominance” hereditary pattern in the Duroc genetic background. In contrast, Black pig and European white pig crossbred displays large variations in coat colors [20]. Therefore, *MC1R* manipulation in the presence of the *dominant white* locus cannot cause a full switch of coat color to black. For coat color manipulation in white pigs, inactivation of the *dominant white* locus is required prior to conferring other desired coat colors by specific gene mutations.

In conclusion, we here present a gene editing approach to realize the coat color switch without affecting gross phenotypes in Duroc or other red pig breeds. This approach can be used to decipher animal coat color genetics precisely and cultivate new animal breeds with desired coat color meeting specific demands in a cost-effective and timesaving way.

## 3. Materials and Methods

### 3.1. MC1R Gene Editing Strategy

To introduce an active *MC1R* into the Duroc genome, we devised an CRISPR/Cas9-mediated HR strategy to precisely replace the Duroc inactive *MC1R* allele (*e*) with the active *MC1R* allele (*E^D1^*) from Chinese indigenous black pigs, with all four missense SNP substitutions (Figure 1B). A gRNA (GGTGTCCAGCCTCTGCTTCC with a TGG PAM site) mediating CRISPR cleavage in the SNP region was designed and ligated into U6-sgRNA cloning vector, as reported previously [21]. The homologous donor was a plasmid containing 2757 bp *MC1R* genome sequence of the *E^D1^* allele covering all four SNPs with approximately 1000 bp flanking homology arms. The gRNA recognized sequence in the donor was mutated to GGTGTCCtctCTgTGtTTtCTcG (mutations are shown with lowercase letters) to avoid CRISPR cleavage at the recombinant *MC1R* locus. (The complete homologous donor sequence was shown in Appendix A.) For HR manipulation, *MC1R* donor plasmid, *MC1R* gRNA plasmid, and *hCas9* plasmid (Addgene # 41815) were co-transfected into Duroc fetal fibroblasts to screen gene-recombinant cells.

### 3.2. MC1R-Modified Fibroblast Screening

The fetal fibroblasts isolated from 30-day Duroc fetuses were primarily cultured and transfected with the CRISPR HR system (three plasmids as previously noted) by nucleofection (Lonza, Basel, Switzerland). The nucleofected cells were seeded in a low density in 10 cm culture dishes (approximately 500 cells/dish). When cells were attached, the medium was replaced with screening medium (high glucose DMEM containing 15% FBS and 500 µg/mL G418 (Thermo Fisher Scientific, Waltham, MA, USA)) for 10 days, with the screening medium refreshed 2–3 times. Following 10 days of screening, the single-cell colonies with compact morphology and suitable size were picked for separate culture in 48-well plates. When cells reached confluence, 1/10 cells in each well were collected for genotyping. The other cells were passaged to 24-well plates. The positive colonies with desired modified *MC1R* alleles were further cultured in 12-well plates and cryopreserved for future use.

### 3.3. Genotyping of MC1R-Modified Fibroblasts

The cell colonies were lysed with proteinase K at 56 °C for 30 min and then the proteinase K was inactivated at 95 °C for 5 min. The cell lysate was directly used as a PCR template to amplify the *MC1R* region using the following primer pair, *MC1R*-genotyping-forward: CCTGCACTCGCCCATGTACTACT, and *MC1R*-genotyping-reverse: GCCAGAAAGAGGTTGACGTT. The primer location in the *MC1R* genomic region is shown in Figure 1B. The PCR product was sequenced using the *MC1R*-genotyping-forward primer. The positive colonies should have homozygous or heterozygous mutation in the four indicated SNPs and no Indel in the gRNA-recognized region.

### 3.4. Pig Cloning

The *MC1R*-modified pigs were generated by SCNT using the positive colonies as donor cells as previously described [21,22]. The reconstructed embryos were transferred into the uterus of estrus-synchronized sow recipients. Each recipient received approximately 200 embryos. The pregnancy of surrogates was monitored monthly by ultrasound, and the cloned piglets were delivered by natural birth.

### 3.5. Genotyping Assay of MC1R-Modified Duroc Pigs

The genomic DNA of cloned pigs were extracted from ear biopsies, and *MC1R* genotypes were PCR-identified using the same method described in Section 3.3. The *E* alleles were determined by the combinations of four SNP genotypes of the *MC1R* gene. To further confirm the origin breed of the cloned pigs, we used genome-wide SNP data for breed tracing [23,24]. Two dark black cloned piglets for test and 66 individuals from four commercial breeds (Pietrain, Duroc, Landrace, and Large White) as reference were sampled and genotyped with PorcineWENS55K SNP Array containing 56,380 SNPs across the whole genome (WENS foodstuff Group). The proportion of ancestries per pig was assessed using the ADMIXTURE 1.22 program and genome-wide SNP data to trace the breed of origin. The number of ancestries (K) was set at 4 [23].

### 3.6. MC1R mRNA Expression in Cloned Pigs

The ear skins of cloned pigs were lysed with TRIzol Reagent (Thermo Fisher Scientific) for total RNA extraction. The total RNA was subjected to reverse transcription with Hifair V one-step RT-gDNA digestion SuperMix (Yeasen Biotechnology, Shanghai, China) to generate cDNA. Real-time qPCR was performed with ChamQ Universal SYBR qPCR Master Mix (Vazyme, Nanjing, China) in a QuantStudio 5 Real-Time PCR System (Thermo Fisher Scientific) using the cDNA as templates and the primer pair, *MC1R*-qF, GCGTCTTCAAGAACGTCAAC, and *MC1R*-qR, ATGGAGTTGCAGATGACGAG, to determine the *MC1R* mRNA expression level. Each sample was run in triplicate, and *MC1R* expression levels were normalized to *GAPDH*. We also amplified the complete mRNA coding region using cDNA as templates for Sanger sequencing to determine the allele-specific expression level of the *MC1R* gene of cloned pigs.

## Figures and Tables

**Figure 1 ijms-23-10356-f001:**
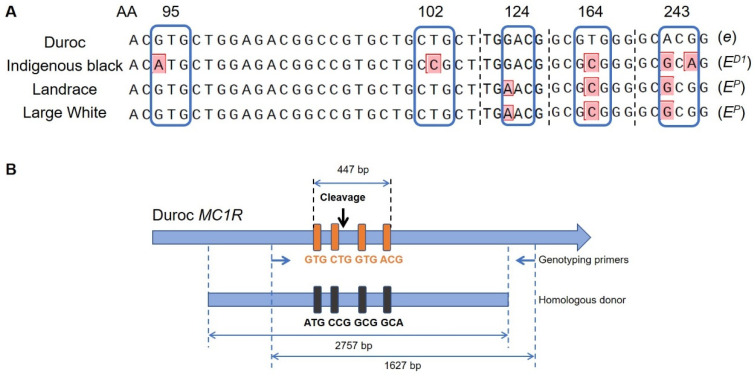
Gene editing strategy at the *MC1R* locus for coat color switch in Duroc pigs. (**A**) *MC1R* sequence alignment among different pig breeds. The four SNPs at AA positions 95, 102, 164, and 243 of the *MC1R* gene distinct between Duroc breed (*e*) and Chinese indigenous black breed (*E^D1^*) are shown in red. Landrace and Large White breeds have a specific D124N substitution compared with other breeds, which is the key variation forming the *E^P^* allele. The *E* allele associated with each *MC1R* variant is indicated in parentheses. (**B**) CRISPR system and donor design for SNP substitutions in Duroc *MC1R*. A guide RNA (gRNA) was designed to recognize the cleavage target in the SNP regions. A 2757-bp DNA sequence was amplified from the *MC1R* genomic region in Chinese indigenous black pigs as the donor to replace the Duroc *MC1R* homologous region by HR. The gRNA-recognized sequence in the donor was synonymously mutated to avoid second CRISPR cleavage after HR. The genotyping primers amplifying a 1627-bp sequence covering homology arm and adjacent genome region after HR were used in PCR assay to detect the occurrence of gene replacement. AA, amino acid position.

**Figure 2 ijms-23-10356-f002:**
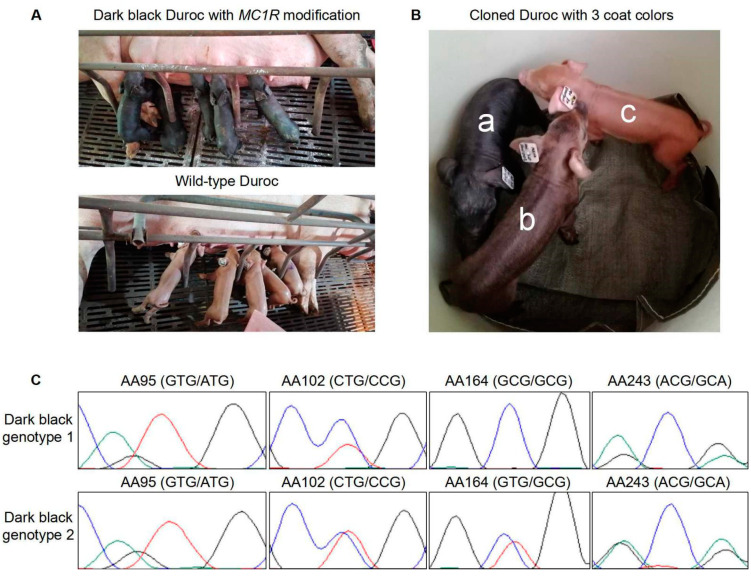
Blackish Durocs generated by *MC1R* gene editing. (**A**) The newborn *MC1R*-modifed Duroc pigs with the dark black coat color (upper panel). As the control, the newborn wild-type Durocs with the red coat color are shown (lower panel). (**B**) The cloned *MC1R*-modified piglets displayed three types of coat color: dark black (a), light black (b), and wild-type red (c). Total substitution in four SNP sites (in one allele or both alleles) resulted in coat color switch in Durocs (a and b). Light black pigs (b) harbored an intact substituted *MC1R* allele, but its expression was lower than the same substituted allele in the dark black pigs (a). Expression level of the substituted *MC1R* allele may decide the black color depth. Non-modification or knockout of the *MC1R e* allele generated the red coat color (c). (**C**) Sanger sequencing results showed that all four SNPs were substituted with corresponding *E^D1^* genotypes on at least one *MC1R* allele in dark black pigs. Among them, nine had the four SNPs substituted on only one allele (dark black genotype 2), and eight had three SNPs substituted on one allele and one SNP substituted on both alleles (dark black genotype 1). AA, amino acid position.

**Figure 3 ijms-23-10356-f003:**
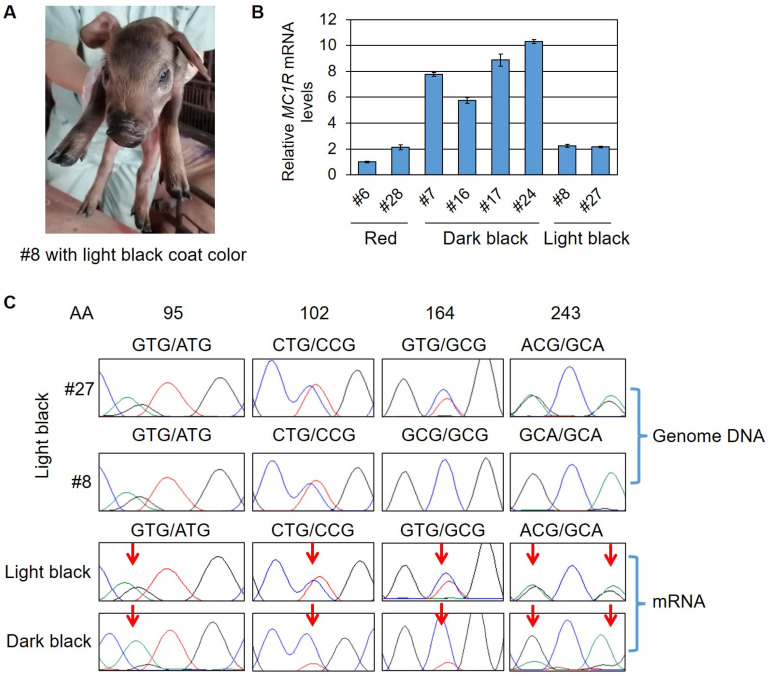
Analysis of *MC1R* gene of cloned Durocs with light black coat color. (**A**) Representative image of a cloned Duroc (#8) showing the light black coat color. (**B**) Quantification of *MC1R* mRNA level in ear skin among cloned Durocs with three different coat colors (red, dark black, and light black). (**C**) Sanger sequencing results of genome DNA showed that #27 had the heterozygous *E^D1^/e* alleles, and #8 had the heterozygous *E^D1^/E^+^* alleles (one allele of *MC1R* was changed to *E^D1^* genotype and the other allele had a V164A substitution forming an *E^+^* genotype). Sanger sequencing of cDNA showed an equal expression level for both *E* alleles in light black pigs, but a great higher expression for the *E^D1^* allele in dark black pigs. Red arrows point to SNP sites varied in *E^D1^* and *e* alleles. AA, amino acid position.

**Figure 4 ijms-23-10356-f004:**
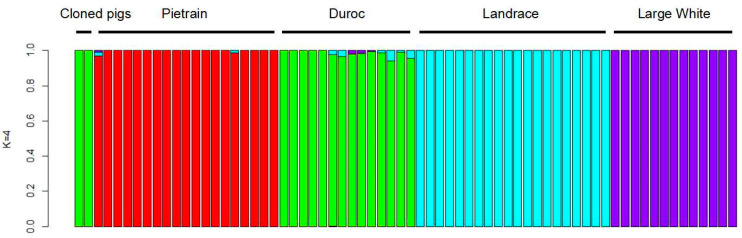
Results of admixture analysis for breed tracing of cloned pigs. Four pig breeds were illustrated with different colors. Each bar represents an individual pig and values on the Y axis represent the proportions of ancestry.

**Table 1 ijms-23-10356-t001:** SCNT summary for *MC1R*-modified pigs.

#Donor Cells	Cell Genotypes ^1^	#Recipient	Embryos Transferred ^2^	Pregnant	Births	Robust at Weaning	Gender	Coat Colors
2-24	Hetero substitutions at 3 SNPs (V95M, L102P, and T243A), homo substitution at V164A	19-937900	205 (164 cleaved)	+	8	5	Male	2 dark black, 1 light black, and 5 red
2-24	Hetero substitutions at 3 SNPs (V95M, L102P, and T243A), homo substitution at V164A	20-694608	206 (169 cleaved)	+	3	1	Male	2 dark black and 1 red
2-24	Hetero substitutions at 3 SNPs (V95M, L102P, and T243A), homo substitution at V164A	20-006896	238 (202 cleaved)	+	6	2	Male	4 dark black and 2 red
8-27	Hetero substitutions at all 4 SNPs	20-627908	207 (166 cleaved)	-	-	-	-	-
8-27	Hetero substitutions at all 4 SNPs	20-688700	204 (181 cleaved)	+	7	6	Female	All dark black
8-27	Hetero substitutions at all 4 SNPs	19-914004	206 (184 cleaved)	-	-	-	-	-
2-24/8-27/8-18 ^3^	2-24, hetero substitutions at 3 SNPs (V95M, L102P, and T243A), homo substitution at V164A; 8-27 and 8-18, hetero substitutions at all 4 SNPs	19-940104	232 (cleaved unknown)	-	-	-	-	-
2-24/8-27/8-18	2-24, hetero substitutions at 3 SNPs (V95M, L102P, and T243A), homo substitution at V164A; 8-27 and 8-18, hetero substitutions at all 4 SNPs	18-419006	247 (cleaved unknown)	+	6	5	Female	2 dark black, 1 light black, and 3 red
2-24/8-27/8-18	2-24, hetero substitutions at 3 SNPs (V95M, L102P, and T243A), homo substitution at V164A; 8-27 and 8-18, hetero substitutions at all 4 SNPs	20-006907	193 (all cleaved)	+	0	0	-	-

^1^ Identified by presence of base substitutions in sequencing chromatograms of *MC1R* gene. ^2^ Cleaved and uncleaved embryos were all transferred into recipients. ^3^ Three cell colonies were mixed as nuclear donors for SCNT.

**Table 2 ijms-23-10356-t002:** Phenotypes and genotypes of cloned piglets.

#Recipient	#Cloned Piglets	Health Status	Coat Color	*MC1R* Genotype					Mutation Types	*E* Allele
				AA95	AA102	gRNA Site ^1^	AA164	AA243		
19-937900	1	Stillborn	Dark black	GTG/ATG	CTG/CCG	Homo sub	GCG/GCG	ACG/GCA	Hetero substitutions at 3 sites, homo substitution at 1 site	*E^D1^/e*
	2	Stillborn	Red	GTG/GTG	CTG/CTG	Δ1/Δ1	GTG/GTG	ACG/ACG	Homo frameshift Indel	*e/e*
	3	Stillborn	Red	GTG/GTG	CTG/CTG	WT/Δ4	GTG/GTG	ACG/ACG	Hetero frameshift Indel	*e/e*
	4	Dead before weaning	Red	GTG/GTG	CTG/CTG	Δ1/Δ1	GTG/GTG	ACG/ACG	Homo frameshift Indel	*e/e*
	5	Survived until weaning	Red	GTG/GTG	CTG/CTG	Δ1/Δ1	GTG/GTG	ACG/ACG	Homo frameshift Indel	*e/e*
	6	Survived until weaning	Red	GTG/GTG	CTG/CTG	WT	GTG/GTG	ACG/ACG	None	*e/e*
	7	Survived until weaning	Dark black	GTG/ATG	CTG/CCG	Homo sub	GCG/GCG	ACG/GCA	Hetero substitutions at 3 sites, homo substitution at 1 site	*E^D1^/e*
	8	Survived until weaning	Light black	GTG/ATG	CTG/CCG	Homo sub	GCG/GCG	GCA/GCA	Hetero substitutions at 2 sites, homo substitution at 2 sites	*E^D1^/E^+^*
20-694608	9	Dead before weaning	Dark black	GTG/ATG	CTG/CCG	Homo sub	GCG/GCG	ACG/GCA	Hetero substitutions at 3 sites, homo substitution at 1 site	*E^D1^/e*
	10	Dead before weaning	Red	GTG/GTG	CTG/CTG	+1/+1	GTG/GTG	ACG/ACG	Homo frameshift Indel	*e/e*
	11	Survived until weaning	Dark black	GTG/ATG	CTG/CCG	Homo sub	GCG/GCG	ACG/GCA	Hetero substitutions at 3 sites, homo substitution at 1 site	*E^D1^/e*
20-006896	12	Mummy	Red	GTG/GTG	CTG/CTG	Δ1/Δ1	GTG/GTG	ACG/ACG	Homo frameshift Indel	*e/e*
	13	Stillborn	Dark black	GTG/ATG	CTG/CCG	Homo sub	GCG/GCG	ACG/GCA	Hetero substitutions at 3 sites, homo substitution at 1 site	*E^D1^/e*
	14	Stillborn	Red	GTG/GTG	CTG/CTG	Δ1/Δ1	GTG/GTG	ACG/ACG	Homo frameshift Indel	*e/e*
	15	Dead before weaning	Dark black	GTG/ATG	CTG/CCG	Homo sub	GCG/GCG	ACG/GCA	Hetero substitutions at 3 sites, homo substitution at 1 site	*E^D1^/e*
	16	Survived until weaning	Dark black	GTG/ATG	CTG/CCG	Homo sub	GCG/GCG	ACG/GCA	Hetero substitutions at 3 sites, homo substitution at 1 site	*E^D1^/e*
	17	Survived until weaning	Dark black	GTG/ATG	CTG/CCG	Homo sub	GCG/GCG	ACG/GCA	Hetero substitutions at 3 sites, homo substitution at 1 site	*E^D1^/e*
20-688700	18	Stillborn	Dark black	GTG/ATG	CTG/CCG	Homo sub	GTG/GCG	ACG/GCA	Hetero substitutions at 4 sites	*E^D1^/e*
	19	Survived until weaning	Dark black	GTG/ATG	CTG/CCG	Homo sub	GTG/GCG	ACG/GCA	Hetero substitutions at 4 sites	*E^D1^/e*
	20	Survived until weaning	Dark black	GTG/ATG	CTG/CCG	Homo sub	GTG/GCG	ACG/GCA	Hetero substitutions at 4 sites	*E^D1^/e*
	21	Survived until weaning	Dark black	GTG/ATG	CTG/CCG	Homo sub	GTG/GCG	ACG/GCA	Hetero substitutions at 4 sites	*E^D1^/e*
	22	Survived until weaning	Dark black	GTG/ATG	CTG/CCG	Homo sub	GTG/GCG	ACG/GCA	Hetero substitutions at 4 sites	*E^D1^/e*
	23	Survived until weaning	Dark black	GTG/ATG	CTG/CCG	Homo sub	GTG/GCG	ACG/GCA	Hetero substitutions at 4 sites	*E^D1^/e*
	24	Survived until weaning	Dark black	GTG/ATG	CTG/CCG	Homo sub	GTG/GCG	ACG/GCA	Hetero substitutions at 4 sites	*E^D1^/e*
18-419006	25	Dead before weaning	Dark black	GTG/ATG	CTG/CCG	Homo sub	GTG/GCG	ACG/GCA	Hetero substitutions at 4 sites	*E^D1^/e*
	26	Survived until weaning	Dark black	GTG/ATG	CTG/CCG	Homo sub	GTG/GCG	ACG/GCA	Hetero substitutions at 4 sites	*E^D1^/e*
	27	Survived until weaning	Light black	GTG/ATG	CTG/CCG	Hetero sub	GTG/GCG	ACG/GCA	Hetero substitutions at 4 sites	*E^D1^/e*
	28	Survived until weaning	Red	GTG/GTG	CTG/CTG	WT	GTG/GTG	ACG/ACG	None	*e/e*
	29	Survived until weaning	Red	GTG/GTG	CTG/CTG	WT	GTG/GTG	ACG/ACG	None	*e/e*
	30	Survived until weaning	Red	GTG/GTG	CTG/CTG	WT	GTG/GTG	ACG/ACG	None	*e/e*

^1^ Mutation types at gRNA site: homo sub, homozygous substitution with synonymous mutation; Hetero sub, heterozygous substitution with synonymous mutation; WT, wild type; Δ1 or +1, Indel size.

## Data Availability

All original data are available upon request.

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
