# Peer review of "Pig Coat Color Manipulation by MC1R Gene Editing"

_ijms, 2022, doi:10.3390/ijms231810356_

Round 1
Reviewer 1 Report
This study conducted by Haiwen Zhong, et al., demonstrated that four missense mutations in the melanocortin receptor 1 gene affect MC1R protein function for eumelanin synthesis. With the aim to devise a genetic modification method for coat color manipulation, they mutated the e allele in the Duroc breed to the dominant ED1 allele using CRISPR-mediated homologous recombination for the four mutation substitutions in the MC1R locus. The MC1R-modified Duroc pigs generated using the allele replacement strategy displayed uniform black coat color across the body. Genotyping assay showed that the MC1R-modified Duroc pigs had a heterozygous ED1/e allele in the MC1R locus, also the pigs remained in the Duroc genetic background. I have a few minor comments and after addressing those comments, I recommend this manuscript for publication in the International Journal of Molecular Sciences.
Comments;
1. Line 33 reference is required.
2. Line 35 reference is required.
3. Line 40 gives appropriate reference.
4. Line 45 reference?
5. Statements in line 48 and line 49 give appropriate references.
6. Line 73 proves the heterozygous substitution in all four SNPs.
7. Melanin was not compared between the control, cloned, homozygous, and heterozygous piglets. Eumelanin (black/brown) and phaeomelanin (yellow/red).
8. Appropriate references are required in the discussion section too.
Author Response
- Line 33 reference is required.
We have added 3 references describing the fat deposition character in Chinese indigenous pig breeds.
“Guo, J.; Shan, T.; Wu, T.; Zhu, L.N.; Ren, Y.; An, S.; Wang, Y. Comparisons of different muscle metabolic enzymes and muscle fiber types in Jinhua and Landrace pigs. J. Anim. Sci. 2011, 89, 185–191.
Liu, X.; Xiong, X.; Yang, J.; Zhou, L.; Yang, B.; Ai, H.; Ma, H.; Xie, X.; Huang, Y.; Fang, S. Genome-wide association analyses for meat quality traits in Chinese Erhualian pigs and a Western Duroc × (Landrace × Yorkshire) commercial population. Genet. Sel. Evol. 2015, 47, 44.
Wu, T.; Zhang, Z.; Yuan, Z.; Lo, L.J.; Chen, J.; Wang, Y.; Peng, J. Distinctive Genes Determine Different Intramuscular Fat and Muscle Fiber Ratios of the longissimus Dorsi Muscles in Jinhua and Landrace Pigs. PLoS ONE 2013, 8, e53181.”
- Line 35 reference is required.
We cannot find some specific academic publications about the prices between Chinese indigenous pig vs Western lean pigs. This is the daily life experience for us (as the Chinese authors) that the pork price of indigenous pigs is higher than western lean pigs. We thus think such statement could be a current common sense thus no need a specific citation.
- Line 40 gives appropriate reference.
A reference about MC1R gene discovery is given here:
“Robbins, L.S.; Nadeau, J.H.; Johnson, K.R.; Kelly, M.A.; Roselli-Rehfuss, L.; Baack, E.; Mountjoy, K.G.; Cone, R.D. Pigmentation phenotypes of variant extension locus alleles result from point mutations that alter MSH receptor function. Cell. 1993, 72, 827-834.”
- Line 45 reference?
The reference discovering the relationship between MC1R genotypes and pig coat color is used here: “Kijas, J.M.; Wales, R.; Tornsten, A.; Chardon, P.; Moller, M.; Andersson, L. Melanocortin receptor 1 (MC1R) mutations and coat color in pigs. Genetics 1998, 150, 1177–1185.”
- Statements in line 48 and line 49 give appropriate references.
We have added two refences for the statements as follows:
“Nishikawa, S.; Kusakabe, M.; Yoshinaga, K.; Ogawa, M.; Hayashi, S.; Kunisada, T.; Era, T.; Sakakura, T.; Nishikawa, S. In utero manipulation of coat color formation by a monoclonal anti-c-kit antibody: two distinct waves of c-kit-dependency during melanocyte development. The EMBO journal 1991, 10, 2111–2118.
Zhang, X.; Li, W.; Liu, C.; Peng, X.; Lin, J.; He, S.; Li, X.; Han, B.; Zhang, N.; Wu, Y.; Chen, L.; Wang, L.; MaYila; Huang, J.; Liu, M. Alteration of sheep coat color pattern by disruption of ASIP gene via CRISPR Cas9. Scientific reports 2017, 7, 8149. ”
- Line 73 proves the heterozygous substitution in all four SNPs.
For cell colony genotyping, we used PCR as described in Materials and methods section 4.3 to amply MC1R gene and then Sanger sequencing to identify the presence of target mutation. Heterozygous substitution in specific nucleotide will display a double peak in sequencing chromatography. The representative sequencing image is similar to the Figure 2C showing cloned positive pig MC1R sequencing result. We thus did not put the cell results in manuscript. Additionally, screened cells are not 100% in their purity in genotype, as the screening process cannot ensure each clone is completely formed by a single cell. Therefore, the cell clone sequencing results can confirm a majority of cells in this clone have the observed genotype per sequencing result, but a minor part of cells with different genotypes may also exist.
- Melanin was not compared between the control, cloned, homozygous, and heterozygous piglets. Eumelanin (black/brown) and phaeomelanin (yellow/red).
Thank you for your comment. As coat color is a clear macro-phenotype that can be easily identified in animal furs, we thus did not conduct microscopic morphology assay to analyze animal coat color. We think the images in Figure 2A and B can better reflect the melanin status in these pigs.
- Appropriate references are required in the discussion section too.
We have rewritten the discussion section to incorporation it with result section, as the discussion section is too short and many discussion contents were included in results in the previous version of manuscript. We also added necessary citations in these sections as follows,
“17. Hammer, R.E.; Pursel, V.G.; Rexroad, C.E.; Jr, Wall, R.J.; Bolt, D.J.; Ebert, K.M.; Palmiter, R.D.; Brinster, R.L. Production of transgenic rabbits, sheep and pigs by microinjection. Nature 1985, 315, 680–683.
- Clark, J.; Whitelaw, B. A future for transgenic livestock. Nat. Rev. Genet. 2003, 4, 825-833.
- Hur, S.J.; Jeong, T.C.; Kim, G.D.; Jeong, J.Y.; Cho, I.C.; Lim, H.T.; Kim, B.W.; Joo, S.T. Comparison of live performance and meat quality parameter of cross bred (korean native black pig and landrace) pigs with different coat colors. Asian-Australasian journal of animal sciences 2013, 26, 1047–1053.”
Reviewer 2 Report
Include the ethical statement
The introduction and Discussion section is quite redundant and distracting, need to rephrase and rewrite
Author Response
We have rewritten the discussion section to incorporation it with result section, as the discussion section is too short and many discussion contents were included in results in the previous version of manuscript. The revised version will be more reasonably structured and less redundant in content.
Reviewer 3 Report
1. As described by the authors, they aim to cultivate a pig breed with high growth rate and meat production as well as a preferred coat color by the gene editing method. In some countries, peoples tend to buy pork with a black coat mainly because they believe that meat from pigs with a black coat color has high flavor and quality. If we choose to improve the growth rate and meat production, a decrease in meat flavor and quality may occur. Does It mean that a pig breed with high growth rate and meat production as well as a preferred coat color may mislead consumers in purchasing pork?
2. The authors described four mutations in the present study. Which mutation locus has the dominant effect in controlling the coat color?
Why did the authors choose to mutate all four SNPs? Maybe one or two SNPs can achieve a higher success rate?
3. Did the editing of the MC1R gene affect other phenotypes, such as viability and growth rate? This should be discussed.
4. Why no homozygous mutants were screened out? Please discuss.
5. The structure of the present manuscript should be improved. In the Results, the authors present their findings and discussed them by citing some references. However, the authors also provided a Discussion section, which is very short. I think that some contents in the Results can be move to the Discussion. Or combined Results and Discussion?
6. The present study chooses MC1R as the target gene. Are there any other genes that participate in regulating the coat color of pigs? Or did the pig coat color is controlled by a single gene or polygenes?
7. Line 84, the four SNPs at AA positions 95, 124......, here should be 95, 102?
8. In figure 3C, #27 and #8 have the same coat color but different genotypes, please discuss.
9. Line 172, #8 and #27 had a similar MC1R expression level with the wild-type Durocs. However, they had different coat colors of light black and red. Should the expression level of MC1R affect coat color?
Author Response
- As described by the authors, they aim to cultivate a pig breed with high growth rate and meat production as well as a preferred coat color by the gene editing method. In some countries, peoples tend to buy pork with a black coat mainly because they believe that meat from pigs with a black coat color has high flavor and quality. If we choose to improve the growth rate and meat production, a decrease in meat flavor and quality may occur. Does It mean that a pig breed with high growth rate and meat production as well as a preferred coat color may mislead consumers in purchasing pork?
Thank you for your good comment. It is true that a large black pigs will lose its pork quality. The two parts between growth trait and meat quality seem hard to coexist. This project was actually inspired from some pig farmers or companies. The current marketing status in China surely favor coat color compared with pork flavor, as flavor is a quite personal feeling that cannot be easily used as standard to classify pork quality. A large black pig can surely bring more profit to farmers or companies currently. This situation could be changed in future, but the present market still has higher price for black pigs in China.
- The authors described four mutations in the present study. Which mutation locus has the dominant effect in controlling the coat color?
Why did the authors choose to mutate all four SNPs? Maybe one or two SNPs can achieve a higher success rate?
For the 4 manipulated mutations (V95M, L102P, V164A, and T243A), at least 3 are necessary to show a full solid black coat color. We have discussed them in the manuscript. V95M and L102P substitutions in the ED1 allele may contribute to the active function of MC1R, thus accounting for the dominant black phenotype. A243T is believed to substantially disrupt MC1R function, resulting in a red coat color phenotype of Duroc pigs. Therefore, the 3 sites (V95M, L102P, and T243A) must have the indicated phenotypes to realize the full solid coat color, or else a light black could occur if less sites are mutated.
- Did the editing of the MC1R gene affect other phenotypes, such as viability and growth rate? This should be discussed.
From our current observation, these modified pigs have no significant abnormality compared with WT Duroc pigs. The long effect for growth and reproduction should be tested in future work. We should note that such modification is not an unnatural genotype. It is an existed gene in many black pig breeds. Thus, our modified pigs may be different from transgenic or gene-editing pigs which harbor non-pig original genes.
- Why no homozygous mutants were screened out? Please discuss.
Because we used a long homologous donor (about 2700bp) for knock-in (KI) manipulation, the KI rates were thus quite low and no homozygous mutants were obtained. We also tried to use a short donor of 500 bp, but the KI rate was even lower that no any mutants can be screened. This may be because a short donor has insufficient homology arm for recombination. The KI rate in pig primary fibroblast is always low from our and many other labs. We have discussed it in the manuscript as follows,
“This result indicates HR remains a low-efficiency event in primary fibroblast, even in the assistance of CRISPR. Previous reports showed CRISPR-mediated HR occurred in an efficiency ranged from 0 to 41.2% in pig fetal fibroblast with a very low rate in most cases, far less than the CRISPR-mediated knockout efficiency”.
- The structure of the present manuscript should be improved. In the Results, the authors present their findings and discussed them by citing some references. However, the authors also provided a Discussion section, which is very short. I think that some contents in the Results can be move to the Discussion. Or combined Results and Discussion?
Thank you for your valuable comment. We have rewritten the discussion section to incorporation it with result section, as the discussion section is too short and many discussion contents were included in results in the previous version of manuscript. The revised version will be more reasonably structured and less redundant in content.
- The present study chooses MC1R as the target gene. Are there any other genes that participate in regulating the coat color of pigs? Or did the pig coat color is controlled by a single gene or polygenes?
Animal coat color is controlled by polygenes. Many genes are involved in regulating animal coat colors or patterns. For example, c-kit is the main locus controlling white coat color in pigs. Our strategy changing red to black only works in Duroc or other red pig breed background. For manipulation of other coat colors or patterns, different genes other than MC1R may be considered. We have discussed this issue in the manuscript.
- Line 84, the four SNPs at AA positions 95, 124......, here should be 95, 102?
Thank you for your careful review. We have revised this mistake in the revised manuscript.
- In figure 3C, #27 and #8 have the same coat color but different genotypes, please discuss.
Yes, this is an important problem we have met in this study. We have carefully investigated it and addressed it in our results. The Figure 3 and its corresponding data is the answer of this question. Although the light black pigs (#27 and #8) harbored the same substituted MC1R allele as the solid black pigs, the MC1R has a low expression level in light black pigs compared with dark black pigs. These results imply that the dominance of ED1 depends on its higher expression than e. An intact ED1 but without sufficient expression level would not support a complete dominant black coat color. For the reason why the ED1 are not expressed sufficiently in the two light black pigs, we anticipated that the recombinant ED1 allele has distal mutations or structural variations caused by the HR process, which impaired ED1 expression. The detailed mechanism remains to be investigated further. These contents have been described in the results and discussion section.
- Line 172, #8 and #27 had similar MC1R expression level with the wild-type Durocs. However, they had different coat colors of light black and red. Should the expression level of MC1R affect coat color?
Yes, as answered in the previous questions, our results imply that a full dominance of ED1 depends on its higher expression than e. An intact ED1 but without sufficient expression level would not support a complete dominant black coat color. This means ED1 with sufficient high expression level shows dark black, ED1 with insufficient expression level shows light black, and e showed red.
Reviewer 4 Report
In this manuscript, Zhong and colleagues generated Duroc pigs with black coat color by MC1R mutation substitutions. This might meet the needs of market in combination with meat quality manipulation. Overall, the results of this manuscript are not clearly presented, and do not sufficiently support the conclusion.
Major Comments:
1. In Fig. 1, the blue box 124 does not indicate D124N, and it might be GAC to AAC.
2. In Fig. 2C, it is confusing that V164A substituted on both allele while other three substitutions were heterozygous, and this situation is quite a lot (8/17). It is not consistent with the design that the four sites are linked in donor. This needs to be explained with further experimental evidence. I suppose the Duroc WT cell line itself has a V164A mutation at this site, so T-vector cloning sequencing is needed to clarify 164(GCG) linked with other three wild genotype SNPs both in WT cell lines and gene-edited pigs. In addition, the four identified SNPs may not be conserved, and experimental evidence need to be provided in multiple cell lines before generating gene-edited pigs. According to Table1 and Table2, V164A mutation may exist in the cell line 2-24. Therefore, V164A may be an unnecessary editing site.
3. In Fig. 3B, the expression levels of e and E alleles should be added.
4. Fig.3C should show 3 kinds of sequencing of mRNA in 3 breeds of pigs (red, light black, dark black).
5. In Fig.3C, the mRNA expression of dark black is completely different from the description, which may be the expression of WT pigs.
Author Response
- In Fig. 1, the blue box 124 does not indicate D124N, and it might be GAC to AAC.
--Thank you for pointing this mistake. We have revise the boxed region in revised figure.
- In Fig. 2C, it is confusing that V164A substituted on both allele while other three substitutions were heterozygous, and this situation is quite a lot (8/17). It is not consistent with the design that the four sites are linked in donor. This needs to be explained with further experimental evidence. I suppose the Duroc WT cell line itself has a V164A mutation at this site, so T-vector cloning sequencing is needed to clarify 164(GCG) linked with other three wild genotype SNPs both in WT cell lines and gene-edited pigs. In addition, the four identified SNPs may not be conserved, and experimental evidence need to be provided in multiple cell lines before generating gene-edited pigs. According to Table1 and Table2, V164A mutation may exist in the cell line 2-24. Therefore, V164A may be an unnecessary editing site.
--Thank you for your comment. The reason for many cells/pigs have homozygous mutation in only V164A site is because V164A is the closest site to CRISPR-cleavage site (80 bp), thus possessing a higher knock-in efficiency. For gene knock-in manipulation, the gene knock-in efficiency is dependent on the distance between the replacement site and the gene cleavage site. For a long donor, the knock-in will only occur in near donor site, while the knock-in rate in distal site is much lower than neat site. This does not mean these sites are not linked, because occurrence of knock-in result in a break in donor.
During cell screening, we screened many cell colonies with partial mutations at only 1 (aa164) or 2 sites (aa102, 164). The major reason is the 2 sites is close to CRISPR cleavage site. In addition, the original fibroblasts used for gene editing have been sequenced in these sites, which showing no presence of these designed substitutions. We have added some sentences to discuss this as follows, “Because AA164 is the closest SNP to the CRISPR-cleavage site (approximately 80 bp) among the four SNPs, AA164 has a much high HR rate than the other three SNPs.”
Furthermore, V164A is actually unrelated to coat color, this point has been confirmed in other publications and we also described this in our manuscript. We mutated it just for an easy construction of donor plasmid, and surely this mutation is existed in black pigs.
- In Fig. 3B, the expression levels of e and E alleles should be added.
--The label “red” is e/e allele and “Dark” is ED1/e allele. For a separated single e or E allele, its expression cannot be distinguished by qPCR in animals.
- 3C should show 3 kinds of sequencing of mRNA in 3 breeds of pigs (red, light black, dark black).
--Thank you for your comment. As mRNA data aim to compare e and E expression, we thus did not include a red pig data which only has e allele. Comparing the overlapped peaks between e and E is a reflection of e and E expression level. Only a single e peak in red pig cannot offer a support data.
- In Fig.3C, the mRNA expression of dark black is completely different from the description, which may be the expression of WT pigs.
--Thank you for your careful review and pointing this critical error. We used wrong sequencing image in assembly of the figure. A cDNA sequencing data from WT Duroc was mistakenly used for dark black pigs. We have carefully checked our raw sequencing data and using a correct sequencing data to replace the original one. Thank you again.
Round 2
Reviewer 4 Report
Please provide relevant experimental data to support your explanation.
Author Response
Dear reviewer,
Thank you for reviewing our manuscripts. To further clarify the high homozygous mutation rate issue, we carefully analyzed the sequencing data and presented data and figures in the attached rebuttal letter to prove our point, that the donor sequences closer to CRISPR cleavage site have the higher gene knock-in efficiency. This point has been demonstrated in many other publications. Please see the attachment for the detailed data addressing this question.
Best Regards,
Huaqiang Yang

Round 3
Reviewer 4 Report
Thank you for your explanation.